# Modulation of α-Mannosidase 8 by Antarctic Endophytic Fungi in Strawberry Plants Under Heat Waves and Water Deficit Stress

**DOI:** 10.3390/ijms262311650

**Published:** 2025-12-01

**Authors:** Daniel Bustos, Luis Morales-Quintana, Gabriela Urra, Francisca Arriaza-Rodríguez, Stephan Pollmann, Angela Méndez-Yáñez, Patricio Ramos

**Affiliations:** 1Laboratorio de Bioinformática y Química Computacional, Departamento de Medicina Traslacional, Facultad de Medicina, Universidad Católica del Maule, Talca 3480094, Chile; dbustos@ucm.cl (D.B.); gurra@ucm.cl (G.U.); 2Multidisciplinary Agroindustry Research Laboratory, Instituto de Ciencias Biomédicas, Facultad de Ciencias de la Salud, Universidad Autónoma de Chile, Cinco Poniente #1670, Talca 3460000, Chile; luis.morales@uautonoma.cl (L.M.-Q.); franciscarodriguezarriaza@gmail.com (F.A.-R.); 3Multidisciplinary Agroindustry Research Laboratory, Instituto de Ciencias Aplicadas, Facultad de Ciencias de la Salud, Universidad Autónoma de Chile, Cinco Poniente #1670 Región del Maule, Talca 3460000, Chile; 4Centro de Biotecnología y Genómica de Plantas, Instituto Nacional de Investigación y Tecnología Agraria y Alimentación (INIA/CSIC), Universidad Politécnica de Madrid (UPM), Campus de Montegancedo, 28223 Madrid, Spain; stephan.pollmann@upm.es; 5Plant Microorganism Interaction Laboratory, Instituto de Ciencias Biológicas, Universidad de Talca, Talca 3460787, Chile

**Keywords:** α-mannosidase, climate change, commercial strawberry, endophytic fungi, heat and water deficit stress

## Abstract

Plant–microbe interactions exert a significant influence on host stress responses; however, the molecular mechanisms underlying these effects remain inadequately understood. In this study, we characterize FaMAN8, an α-mannosidase from *Fragaria × ananassa*, to explore its role in adaptation to heat waves and water deficit, as well as its modulation by fungal endophytes. Transcriptomic analysis identified *FaMAN8* as the sole α-mannosidase isoform highly conserved across reported sequences, with root-specific induction under conditions of heat stress, deficient irrigation, and endophytic colonization. Structural modeling revealed that FaMAN8 exhibits the canonical domain organization of glycoside hydrolase family 38 (GH38) enzymes, featuring a conserved catalytic architecture and metal-binding site. Molecular docking and dynamics simulations with the Man_3_GlcNAc_2_ ligand indicated a stable binding pocket involving key catalytic residues and strong electrostatic complementarity. MM-GBSA and free energy landscape analyses further supported the thermodynamic stability of the protein–ligand complex. Cavity analysis revealed a larger active site in FaMAN8 compared to its homolog JbMAN, suggesting broader substrate accommodation. Collectively, these findings identify FaMAN8 as a stress-responsive glycosidase potentially involved in glycan remodeling during beneficial root–fungus interactions. This work provides molecular insights into plant–microbe symbiosis and lays the groundwork for microbiome-informed strategies to enhance crop stress resilience.

## 1. Introduction

The current climate change scenario, coupled with future projections of extreme weather conditions globally, such as heatwaves, water scarcity, and inland flooding, necessitates the implementation of strategies that facilitate organisms’ adaptation to new environmental conditions [1]. Agriculture is among the most adversely impacted economic sectors by these changes, which result in crop damage and a substantial decline in agricultural productivity [2]. Consequently, food security is jeopardized, complicating efforts to achieve the second Sustainable Development Goal of the 2030 Agenda, which seeks to eradicate hunger. The United Nations Framework Convention on Climate Change strongly emphasizes that climate change is an existing reality. Therefore, it is imperative to develop adaptive and resilient solutions that integrate social, economic, and ecological dimensions, addressing both current impacts and those anticipated in the medium and long term [3].

Various strategies have been implemented to enhance plant adaptability in response to emerging environmental scenarios, including genetic reprogramming, genetic breeding, and the interaction with symbiotic organisms [4,5]. Over the past decade, endophytic fungi have been extensively studied; the process by which symbiosis is established between endophytic fungi and host plants has been described by Li et al. (2025) [6]. Upon establishment of this interaction, the immune response is activated by elicitors produced by the endophytic fungi, which bind to key plant-specific molecules, thereby triggering a wide range of responses [7,8,9]. For example, under conditions of environmental stress, endophytic fungi within agricultural systems offer numerous advantages. These include the biosynthesis of specific phytohormones, the indirect promotion of the expression of key stress tolerance and resistance genes, the enhancement of antioxidant enzyme activity, and the facilitation of nutrient assimilation from the soil [6,10]. Giauque et al. (2019) [11] propose that the interaction between plants and endophytic fungi is governed by three primary paradigms: evolutionary history, habitat adaptation, and physiological traits. Furthermore, the interaction of endophytic fungi with plants from extreme habitats enhances abiotic stress tolerance and biomass production. In this regard, endophytic fungi from Antarctica, through their symbiotic relationships with vegetable crops or fruit plants, have been shown to confer tolerance to cold, drought, heat, and salt stress [12,13,14,15,16,17,18,19,20]. Notwithstanding these advancements, it remains crucial to ensure that fruits and vegetables cultivated through the interaction between plants and endophytic fungi retain appropriate organoleptic properties—such as color, aroma, and particularly texture, as these attributes are fundamental to consumer acceptance [21,22]. The quality of texture is fundamentally associated with the chemical and structural composition of the cell wall, with particular emphasis on the glycans that comprise it and the complexity of their interactions within this intricate polymeric network [23]. The enzymatic activity responsible for cleaving these polysaccharides, in conjunction with N-glycosylation processes, is crucial for elucidating the primary factors involved in the disassembly and solubilization of the cell wall [24,25,26,27]. Considerable research has been undertaken to investigate the composition of the cell wall and the enzymatic processes involved in the degradation of fruit cell walls [28,29,30,31]. Class I α-mannosidases are integral to the initial stages of N-glycan processing and contribute significantly to root development and cell wall biosynthesis in Arabidopsis. This underscores the pivotal role of these enzymes in both structural and developmental processes [32].

Exoglycosidases play an indirect role in the disassembly of cell walls by removing N-glycosylation residues from enzymes involved in polysaccharide degradation. The removal of sugar moieties, generally obtained as post-translational modifications, can influence enzymatic activity or alter the three-dimensional structure of these proteins. β-d-N-acetylhexosaminidase and α-mannosidase are two exoglycosidases implicated in cell wall remodeling processes associated with fruit ripening [33]; changes in their gene expression levels and enzymatic activity have been documented during this process [34,35,36]. It is particularly intriguing to explore how cell wall remodeling occurs not only within fruit tissue but also throughout the entire plant under abiotic stress conditions. As a sessile organism, the plant must rapidly adapt to stress conditions to ensure its survival. α-Mannosidase has been associated with abiotic stress tolerance in *Triticum aestivum*, where its relative expression is induced under salt, drought, abscisic acid, and H_2_O_2_ stress [37]. Furthermore, under water stress conditions, a downregulation in the differential expression of α-mannosidase has been observed in *Capsicum annuum* L. [38]. Ghosh & Xu (2014) [39] provided a summary of the role of α-mannosidase in plant root responses to abiotic stress, specifically in soybean and watermelon under drought conditions. Endophytic fungi can enhance plant resilience to abiotic stress conditions by upregulating gene expression levels, facilitating the accumulation of specific proteins, and augmenting the activity of key structural proteins and enzymes such as aquaporins, dehydrins, alcohol acyltransferase, and pyruvate decarboxylase [17,22]. However, there is limited information regarding cell wall remodeling in vegetative tissues under abiotic stress conditions, and α-mannosidase activity under abiotic stress remains largely unexplored in fruit or vegetable crops inoculated with endophytic fungi.

In this study, we evaluate the gene expression and structural characterization of an α-mannosidase protein in the vegetative tissues of strawberries, with roots inoculated with endophytic fungi isolated from Antarctic plants (*Penicillium chrysogenum* and *P. brevicompactum*) [17], under simulated abiotic stress conditions of water and heat stress, as a consequence of climate change.

## 2. Results

To identify α-mannosidase genes potentially involved in root–microbe interactions in *F. × ananassa* during drought and heat stress, we conducted a comprehensive in *silico* screening using the transcriptome dataset previously published by Yáñez et al. (2025) [17]. Ten full-length α-mannosidase sequences previously described by Méndez-Yáñez et al. (2024) [28] were used as query templates in local BLASTn searches against the assembled transcriptome. Remarkably, only one transcript exhibited perfect sequence identity (100%) with one of the known α-mannosidases, specifically the α-mannosidase from *F. × ananassa* (FxaC_21g01481.t1). Based on this unambiguous match, this transcript was selected for further structural and functional characterization. The exclusive identification of *FaMAN8* among the ten reference sequences suggests that this isoform may represent the predominant or possibly the sole α-mannosidase transcript expressed under the specific experimental conditions captured in the Yáñez et al. (2025) [17] dataset. This finding underscores the potential biological significance of FaMAN8 in stress response, as mediated by endophytic fungi. Given the functional diversity within the α-mannosidase gene family, pinpointing a single expressed isoform provides a valuable opportunity to explore specific enzymatic roles *in planta*, particularly in the context of environmental and stress responses mediated by symbiotic interactions.

Subsequent analyses focused on the investigation of the expression pattern, protein structure prediction, and the potential regulatory function of FaMAN8, with particular emphasis on its response to endophytic fungal interaction with the plant under conditions of drought and heat stress.

### 2.1. Expression of FaMAN8 Is Modulated by Endophytic Fungal Interaction

To corroborate the induction of *FaMAN8* as estimated from normalized read counts in RNA-seq data, the transcript abundance of *FaMAN8* was quantified in the roots and leaves of strawberry plants subjected to varying irrigation regimes (100% irrigation (W+), and 50% irrigation (W−)), with or without endophytic inoculation (E+ or E−), and exposed (H+) or not exposed (H−) to heat stress. In roots, *FaMAN8* expression exhibited a pronounced response to the combined presence of full irrigation and endophytic colonization (W+E+), achieving a 4.7-fold increase relative to control conditions (W+E−, H+) (Figure 1). This significant upregulation suggests a synergistic effect between optimal water availability and the presence of endophytes under thermal stress. In contrast, all other root treatments, including those with reduced irrigation (W−), demonstrated markedly lower expression levels, with the lowest transcript accumulation observed in the W−E+ condition under no heat (H−), where *FaMAN8* was nearly undetectable (Figure 1). In leaves, *FaMAN8* transcript levels remained generally low across all treatments, showing no significant upregulation in response to either irrigation regime or endophytic presence. The highest expression was recorded in the W+E− condition under heat stress (H+), yet this value did not differ significantly from most other treatments, indicating that *FaMAN8* is not strongly expressed, or regulated, in foliar tissue under the tested conditions (Figure 1). Collectively, these findings demonstrate a root-specific and highly condition-dependent expression pattern for *FaMAN8*, suggesting that its transcription is tightly regulated by both abiotic (water and temperature) and biotic (endophytic colonization) factors. The strong induction under E+W+H+ conditions may indicate a functional role of FaMAN8 in root adaptation or remodeling processes that support beneficial microbial interactions under environmental stress (Figure 1).

### 2.2. Structural and Domain Organization of FaMAN8 Reveals Structural Conservation with Jack Bean α-Mannosidase

To elucidate the structural characteristics of the deduced FaMAN8 protein, a domain architecture analysis was conducted utilizing the Pfam and SMART databases. The predicted protein sequence revealed a modular structure comprising three conserved domains: an N-terminal polysaccharide deacetylase-like domain, a central α-mannosidase domain, and a C-terminal glycosyl hydrolase family 38 (GH38) domain (Appendix A). To evaluate evolutionary conservation and infer potential functional similarity, the domain organization of FaMAN8 was compared with the well-characterized α-mannosidase from Jack bean (JbMAN) [40]. Both proteins exhibited a highly similar architecture, with comparable positioning and size of the three domains, suggesting that FaMAN8 may share similar enzymatic functions and substrate specificity with JbMAN. To further investigate the structural features of the FaMAN8 protein, a three-dimensional model was generated based on homology modeling, using the well-characterized JbMAN structure as a template (Figure 2). The resulting model revealed a multi-domain organization consistent with the Pfam domain annotation: the N-terminal polysaccharide deacetylase-like domain (pink), the α-mannosidase middle domain (yellow), and the C-terminal glycosyl hydrolase family 38 (GH38) domain (blue) (Figure 2). These domains are spatially organized in a compact fold, forming a plausible catalytic pocket at the interface between the middle and GH38 domains.

The model predicts that FaMAN8 forms a homodimer, as depicted in two orientations (top and bottom panels in Figure 2). Remarkably, each monomer features a conserved metal-binding pocket that accommodates a Zn^2+^ ion (green spheres), coordinated within the catalytic core. Zinc coordination is characteristic of GH38 α-mannosidases and is crucial for catalytic activity, thereby affirming the functional validity of the model. The spatial arrangement of the metal-binding residues aligns with known α-mannosidase structures and supports the hypothesis that FaMAN8 functions via a metal-dependent hydrolytic mechanism (Figure 2). Furthermore, the active site cavity of FaMAN8 demonstrated a substantially larger volume (575.0 ± 791.1 Å^3^) compared to the template JbMAN (PDB: 6B9O, 211.0 ± 133.7 Å^3^), along with an increased surface area (Table 1). This enlargement suggests a more accessible catalytic pocket in FaMAN8, potentially accommodating larger or more complex glycan substrates.

### 2.3. Protein-Ligand Evaluation Suggests Conserved Substrate-Binding Pocket in FaMAN8

To further elucidate the potential enzymatic mechanism of FaMAN8, molecular docking simulations were conducted utilizing the oligomannosidic ligand Man_3_GlcNAc_2_, which serves as a canonical trimming intermediate in N-glycan processing, as delineated by Wang et al. (2020) [37]. The analysis of molecular dynamics (MD) simulations indicated that this ligand is positioned within a deep cleft at the interface of the central and C-terminal domains, closely aligning with the predicted catalytic site (Figure 3).

In the proposed docked model, Man_3_GlcNAc_2_ (depicted as blue spheres) engages in multiple interactions with residues located within the conserved glycosidase groove (Figure 4). These interactions include potential hydrogen bonds and van der Waals contacts with amino acids from both the α-mannosidase and GH38 domains. Notably, the ligand is situated in close proximity to the Zn^2+^ ion (green sphere), which is likely instrumental in catalysis, either by stabilizing the transition state or by polarizing the glycosidic bond during hydrolysis (Figure 3). Furthermore, trajectory analyses demonstrated that both monomers sustained persistent interactions with the ligand throughout the simulation, indicating stable binding modes within the active sites of each chain (Figure 4).

In monomer A (left panel, Figure 4), a consistent interaction profile was identified for residues ASP48, GLN67, ASP168, ARG193, GLU234, ASP291, ASP410, SER815, ARG816, and GLY817. Notably, GLU234 and ASP291 exhibited the highest interaction fractions, primarily through water-mediated contacts (blue bars), with a significant contribution from hydrogen bonding (green bars), suggesting their central role in substrate stabilization and catalysis. In monomer B (right panel, Figure 4), the ligand engaged a partially overlapping yet distinct subset of residues, including ASP48, TRP51, GLN67, ARG193, GLU234, ASP291, ARG351, ASP410, SER815, ARG816, and GLY817. Once again, ASP291 and GLY817 demonstrated the highest interaction occupancy, predominantly through water bridges. Interestingly, TRP51, unique to monomer B, contributed hydrophobic contacts (purple bar), indicating subtle conformational differences between monomers that may influence binding dynamics or allosteric behavior.

To further assess the thermodynamic stability of the FaMAN8–ligand complex, we constructed the free energy landscape (FEL) utilizing the root-mean-square deviation (RMSD) and buried surface area (BSA) as collective variables (CVs) (Figure 5). The FEL reveals a predominant global minimum centered around an RMSD of approximately 3.6 nm and a BSA near 0.0, indicating that the system converged into a stable conformational state during the molecular dynamics simulation. This principal energy basin is characterized by a low free energy range between 0.0–3.0 kJ/mol, highlighted in purple, green, and orange in the landscape (Figure 5). The narrow dispersion of low-energy states suggests a highly populated and energetically favorable conformation of the complex, consistent with stable binding. Minor local minima at higher RMSD and negative BSA values may correspond to transient conformational fluctuations or minor rearrangements at the binding interface; however, these are separated by higher energy barriers (>12 kJ/mol), indicating a limited probability of transition.

Furthermore, the MM-GBSA calculations revealed a total binding free energy profile indicative of a stable FaMAN8–ligand complex (Table 2). The primary contributors to the favorable interaction energy were Coulombic interactions (–40.4 ± 17.8 kcal/mol) and van der Waals forces (–35.8 ± 15.0 kcal/mol), underscoring the strong electrostatic and steric complementarity between the ligand and the binding pocket (Table 2). Hydrogen bonding also provided a modest contribution (–3.9 ± 1.8 kcal/mol), while lipophobic interactions offered additional stabilization (–10.3 ± 4.9 kcal/mol). As anticipated, polar solvation energy was unfavorable (+81.7 ± 30.9 kcal/mol); however, this was counterbalanced by the cumulative stabilizing factors, thereby supporting the overall thermodynamic feasibility of ligand binding.

### 2.4. Electrostatic Surface Potential Evaluations

To explore the electrostatic characteristics of the ligand-binding region in FaMAN8, surface electrostatic potential maps were generated for the homodimeric protein in both ligand-free and ligand-bound states (Figure 6). In Figure 6A, the complete FaMAN8 dimer is depicted, revealing a heterogeneous electrostatic surface characterized by regions of strong negative potential (red) interspersed with positively charged patches (blue). Notably, one monomer exhibits a prominent negatively charged cavity near the catalytic pocket. A closer examination of the monomeric structure (Figure 6B) highlights this negatively charged binding cleft, which supports a favorable electrostatic environment for accommodating carbohydrate ligands such as Man_3_GlcNAc_2_, which possess multiple hydroxyl and amide groups. Upon docking of the ligand (Figure 6C), the binding site remains predominantly negatively charged, with the ligand (green spheres) fitting into the central pocket. The distribution of charges around the binding interface suggests a balance between electrostatic complementarity and polar interactions, particularly with conserved acidic residues such as ASP and GLU identified in previous interactions (Figure 4) and MM-GBSA analyses (Table 2).

## 3. Discussion

Plant–microbe symbioses are increasingly acknowledged as fundamental to plant adaptation under environmental stress conditions. Fungal endophytes, in particular, have been demonstrated to enhance plant tolerance to drought, salinity, and heat by modulating host physiology and gene expression. Within this context, glycosyl hydrolases, such as α-mannosidases, play crucial roles in protein quality control, glycoprotein remodeling, and cell wall modification—processes essential for maintaining cellular homeostasis under stress [17,22,41]. In the present study, we identified and characterized FaMAN8, an α-mannosidase from *F. × ananassa*, to investigate its potential function in root stress physiology and endophyte-mediated responses.

Transcriptomic analysis has identified *FaMAN8* as the only α-mannosidase isoform that is highly conserved among previously documented sequences, indicating a potentially predominant or specialized function in strawberries. Expression analysis revealed that *FaMAN8* is predominantly expressed in roots, with transcript levels significantly increasing under conditions of full irrigation combined with endophytic colonization and heat stress (W+E+H+). This expression pattern suggests a regulation that is context-dependent, influenced by both abiotic and biotic factors, and underscores a potential role for FaMAN8 in stress adaptation and symbiotic signaling [17,22], possibly modulating root development under drought stress [32].

Structural modeling of FaMAN8 has revealed a conserved three-domain architecture characteristic of GH38 α-mannosidases, consisting of an N-terminal polysaccharide deacetylase-like domain, a central α-mannosidase middle domain, and a C-terminal GH38 catalytic domain. The predicted three-dimensional structure closely resembles the template enzyme from *Canavalia ensiformis* (JbMAN), with well-aligned catalytic residues and metal-binding motifs [39]. The dimeric configuration observed in FaMAN8 further supports its enzymatic functionality, as dimerization is often linked to catalytic efficiency and substrate affinity in glycosidases.

Molecular docking with the canonical N-glycan ligand Man_3_GlcNAc_2_ revealed that the ligand precisely fits into a negatively charged catalytic pocket coordinated by Zn^2+^ ions. The interaction network comprised residues GLU234, ASP291, ARG816, and GLY817, which maintained stable hydrogen bonds and water-mediated interactions throughout the simulation. These residues correspond to catalytically relevant positions identified in other GH38 enzymes, suggesting an evolutionary conservation of the catalytic mechanism.

The docking results, which have been corroborated by molecular dynamics simulations, indicated a robust and stable protein–ligand complex. Both monomers of the dimer contributed to substrate stabilization, with slight conformational differences that may confer flexibility in substrate recognition. The MM-GBSA energy decomposition corroborated these findings, indicating that Coulombic and van der Waals interactions were the major contributors to binding stability, while polar solvation energy represented the principal destabilizing factor. The overall negative binding free energy confirmed a thermodynamically favorable interaction between FaMAN8 and Man_3_GlcNAc_2_.

Moreover, the free energy landscape analysis revealed a single dominant conformational basin, suggesting a stable and energetically favorable configuration during the simulation trajectory. Notably, cavity analysis revealed that FaMAN8 possesses a significantly larger active-site pocket compared to JbMAN [40]. This structural feature may allow FaMAN8 to accommodate more complex or branched substrates, potentially broadening its functional scope in vivo. The electrostatic surface analysis further supported this hypothesis, showing an extensive negatively charged groove suitable for interaction with polar carbohydrate ligands. Such physicochemical adaptability could be advantageous under environmental stress, where changes in cell wall composition and glycoprotein turnover are critical for maintaining root integrity and communication with endophytic partners.

These findings collectively indicate that FaMAN8 is a structurally conserved yet functionally adaptive α-mannosidase, potentially involved in glycoprotein remodeling or stress-related cell wall modifications during beneficial plant–fungus interactions, thereby modulating root development to cope with abiotic stress. The integration of in silico modeling, molecular dynamics, and expression profiling offers substantial evidence supporting its role in stress physiology. Further functional validation through gene silencing or overexpression in endophyte-associated strawberry plants is crucial to elucidate the precise biological function of FaMAN8 and its potential as a molecular target for enhancing crop resilience.

## 4. Materials and Methods

### 4.1. Plant and Fungal Material

Plant and fungal materials were sourced from the sample pool previously delineated in the study by Yáñez et al. (2025) [17]. In summary, 400 *Fragaria × ananassa* cv. ‘Aromas’ plants were cultivated in 1 L pots containing a standard peat–perlite substrate (1:1) under greenhouse conditions at the Universidad de Talca, Maule Region, Chile. To produce endophyte-free plants (E−), all specimens were initially treated with the commercial fungicide Benlate^®^ (2 g L^−1^) followed by the broad-spectrum antibiotic rifampicin (50 μg mL^−1^) to eradicate endophytic bacteria. Endophyte-inoculated plants (E+) were generated using two fungal strains previously isolated from *Colobanthus quitensis* and *Deschampsia antarctica*, collected near the Henryk Arctowski Polish station in Admiralty Bay, King George Island, Antarctic Peninsula. Fungal identities were confirmed via ITS sequencing as *Penicillium chrysogenum* (GenBank accession no. KJ881371) and *Penicillium brevicompactum* (GenBank accession no. KJ881370). A 1:1 spore mixture (1 × 10^7^ spores mL^−1^) was prepared and applied directly to the rhizosphere, with a subsequent application conducted 15 days later to ensure effective colonization. To verify the presence or absence of endophytes, root samples were analyzed before and after the experiment using trypan blue staining and light microscopy. Additionally, fungal identity was confirmed by ITS sequencing of axenic cultures re-isolated from inoculated roots.

### 4.2. Identification of α-Mannosidase Transcripts in the Strawberry Transcriptome

To identify potential α-mannosidase-encoding transcripts within the strawberry transcriptome dataset previously published by Yáñez et al. (2025) [17], a comparative sequence analysis was conducted. Specifically, ten full-length α-mannosidase gene sequences, previously characterized in *F. × ananassa* by Méndez-Yáñez et al. (2024) [28], were employed as queries in local BLASTn searches against the assembled transcriptome data. All sequence alignments were performed using BLAST+ (v2.13.0) with default parameters. Among the queried sequences, only one transcript demonstrated a high-confidence match, exhibiting 100% sequence identity with the previously described FaMAN8 protein (FxaC_21g01481.t1). This transcript was designated as FaMAN8 for subsequent analyses.

### 4.3. RNA Extraction and cDNA Synthesis

Total RNA was extracted from 100 mg of root tissue utilizing the Spectrum™ Plant Total RNA Kit (Sigma-Aldrich, St. Louis, MO, USA), in accordance with the manufacturer’s protocol. Residual genomic DNA was eliminated through treatment with RQ1 RNase-free DNase (Promega, Madison, WI, USA), following the supplier’s instructions. To verify the absence of DNA contamination, control qPCR reactions were conducted using RNA samples that had not undergone reverse transcription. The quantity and purity of RNA were assessed using a NanoDrop ND-1000 spectrophotometer (ThermoFisher Scientific, Wilmington, DE, USA). RNA integrity was evaluated via agarose gel electrophoresis. Three independent RNA isolations were performed for each experimental condition. First-strand cDNA was synthesized from 1 µg of total RNA using the First Strand cDNA Synthesis Kit (ThermoFisher Scientific), in accordance with the manufacturer’s recommendations.

### 4.4. Quantitative Real-Time PCR (qPCR) Analysis

Gene expression levels were quantified utilizing quantitative real-time PCR (qPCR) with the PowerUp™ SYBR™ Green Master Mix (ThermoFisher Scientific, Wilmington, DE, USA) on an AriaMx Real-Time PCR System (Agilent Technologies, Santa Clara, CA, USA). Each 20 μL reaction mixture comprised 1 μL of each primer, 1.5 μL of cDNA, 10 μL of SYBR Green Master Mix, and 7 μL of nuclease-free water. The thermal cycling protocol consisted of an initial denaturation at 95 °C for 3 min, followed by 40 amplification cycles at 95 °C for 30 s and 60 °C for 45 s. Relative gene expression was calculated using the 2^−ΔΔCt^ method. Three independent biological replicates were analyzed for each condition. Primers for the target gene were designed based on sequences reported by Méndez-Yáñez et al. (2024) [28], and normalization was conducted using *FaGAPDH* as the internal reference gene, in accordance with previous studies [42,43].

### 4.5. Structural and Bioinformatics Characterization of FaMAN8

#### 4.5.1. Protein Structure Modelling

The three-dimensional structure of the FaMAN8 protein was modeled using homology modeling via the SWISS-MODEL server (https://swissmodel.expasy.org/, accessed on 18 March 2025) [44]. The amino acid sequence of FaMAN8, previously determined, served as the input for this model construction. The crystallographic structure of a closely related homolog, available in the Protein Data Bank (PDB) [45] under accession code 6B9O [40], was chosen as the structural template based on sequence similarity and alignment quality. Given that SWISS-MODEL generates a model for a single polypeptide chain corresponding to the input sequence, this chain was duplicated, and each copy was structurally aligned with the respective chains in the 6B9O template to reconstruct the complete multimeric assembly of FaMAN8. Furthermore, the positions of the zinc cofactors were extracted from the 6B9O structure and incorporated into the modeled FaMAN8 protein to maintain the catalytic environment. The resulting model underwent quality assessment using the validation tools integrated within SWISS-MODEL to ensure its structural reliability and suitability for subsequent computational analyses.

#### 4.5.2. Protein Preparation

Both the crystallographic structure 6B9O, obtained from the PDB, and the newly modeled FaMAN8 were prepared and optimized utilizing the Protein Preparation Wizard within the Maestro/Schrödinger Suite (Schrödinger, LLC, New York, NY, USA) (MAESTRO, 2021) [46]. The preparation protocol included the assignment of bond orders, the addition of missing side chains and loop regions, the optimization of the hydrogen-bonding network, and the adjustment of protonation states to reflect a pH of 5.0. During the energy minimization phase, distinct strategies were employed for each protein. For the 6B9O structure, minimization was restricted to hydrogen atoms only, ensuring proper relaxation of proton positions without altering the heavy-atom backbone. Conversely, for the FaMAN8 model, a restrained minimization of the entire protein was conducted to alleviate local conformational strain while preserving the overall tertiary structure.

#### 4.5.3. Ligand Conformation

In parallel, mannose was selected as the reference ligand based on previously reported studies by Wang et al. (2021) [47]. The ligand was constructed according to the structural information described in the literature and subsequently prepared using the LigPrep module within the Maestro/Schrödinger Suite (Schrödinger, LLC, New York, NY, USA). This process included geometry optimization and protonation state adjustment at pH 5.0. The selection of mannose was consistent with the enzymatic activity of FaMAN8, an exoglycosidase that cleaves terminal α-1,2, α-1,3, and α-1,6 mannose residues from N-glycans, and it was therefore employed as the substrate for all subsequent computational analyses.

#### 4.5.4. Ligand Docking

The synthesized mannose ligand was subjected to docking within the binding sites of both the FaMAN8 model and the 6B9O structure utilizing the Glide module [48] of the Maestro/Schrödinger Suite (version 2020-3) in standard precision (SP) mode. The docking grid was centered on the zinc cofactor, which, as documented in the literature, resides within the active site cavity and delineates the substrate-binding pocket as reported by Howard et al. (2018) [40]. The grid was further expanded to encompass surrounding residues that constitute the active site, employing a cubic box of 30 Å per axis to ensure exhaustive sampling of potential ligand conformations.

During the docking process, the ligand was treated as flexible, permitting conformational exploration within the active site. Up to 10 binding poses were generated and ranked based on the GlideScore, with the highest-scoring conformation being selected as the most likely binding mode for subsequent computational analyses. Given that both the FaMAN8 model and the 6B9O structure consist of two polypeptide chains, this docking procedure was conducted independently for each chain to account for potential conformational and microenvironmental variations across the enzyme subunits.

#### 4.5.5. System Preparation and Molecular Dynamics Preparation

Each protein system was solvated utilizing the System Builder tool within the Maestro/Schrödinger Suite (Schrödinger, LLC, New York, NY, USA). A periodic orthorhombic water box was constructed around each protein employing the SPC water model, with box dimensions specified as 20 × 18 × 18 Å along the x, y, and z axes, respectively. A total of 38 Na^+^ counterions were incorporated to neutralize the net charge of the system. This solvated and charge-balanced configuration served as the initial setup for subsequent molecular dynamics simulations.

Each fully prepared system was imported into the Desmond molecular dynamics engine within the Maestro/Schrödinger Suite (Schrödinger, LLC, New York, NY, USA) and OPLS3e to conduct production simulations (Roos et al., 2019) [49]. Simulations were executed for a total duration of 300 ns under isothermal–isobaric (NPT) ensemble conditions at a constant temperature of 310 K and a pressure of 1.0 bar. Approximately 600 frames were recorded for each simulation, providing sufficient temporal resolution for subsequent analyses. The pressure coupling was maintained with a relaxation time of 500 ps, ensuring system stability and accurate sampling of conformational space. To enhance statistical robustness and account for stochastic variability, the trajectory seed parameter in the advanced simulation settings was adjusted to generate three independent replicates for each prepared system. Based on the methodology described by Elhiti et al. (2012) [50], the structural domains of the FaMAN8 protein were identified and visually distinguished by color. This domain-specific coloring facilitated the interpretation of the protein architecture, allowing clear differentiation of functional regions and supporting the analysis of domain-specific interactions with the mannose ligand.

#### 4.5.6. Computational Analysis

Upon completion of the molecular dynamics simulations, the Simulation Interaction Diagram (SID) module within the Maestro/Schrödinger Suite (Schrödinger, LLC) was employed to examine the interactions between each protein chain and its associated mannose ligand. Interaction data, encompassing hydrogen bonds, salt bridges, and hydrophobic contacts, were extracted for each frame of the trajectory. These data were subsequently processed using a custom R script to compute the interaction fraction for each protein-ligand pair throughout the simulation. This analysis yielded a table identifying conserved residues that maintained interactions with the ligand in over 30% of the trajectory, thereby providing a quantitative assessment of persistent and potentially functionally significant contacts. To assess the conformational and energetic properties of the protein–ligand systems, the solvent-accessible surface area (SASA) was calculated for each individual ligand and protein chain using the Maestro/Schrödinger Suite (Schrödinger, LLC), as well as for the corresponding protein–ligand complexes. The buried surface area (BSA) of each complex was determined using the formula: BSA = Complex − (Protein + Ligand). Concurrently, root-mean-square deviations (RMSD) were calculated for each protein chain and its associated ligand along the trajectory. Notably, only residues identified as conserved from the Interaction Fraction analysis, those maintaining contacts with the ligand in more than 30% of frames across the three replicates, were considered in the RMSD calculations. Two RMSD measurements were conducted per replicate, one for each chain, utilizing the respective subset of relevant residues to capture the most significant conformational dynamics involved in ligand binding.

The quantitative metrics, including the buried surface area (BSA) and root-mean-square deviation (RMSD) of selected residues, were integrated to construct the Free Energy Landscape (FEL; https://zenodo.org/records/10850229, accessed on 25 March 2025) of the protein–ligand systems. This methodology offers a robust characterization of the conformational space and energetically favorable states of protein–ligand interactions, emphasizing the contributions of the most functionally relevant residues to binding stability and dynamics. By plotting free energy as a function of selected coordinates, such as RMSD versus BSA, the FEL elucidates minima corresponding to stable conformational states and barriers representing transitions. This visualization provides both qualitative and quantitative insights into the system’s conformational preferences, facilitating the identification of energetically favorable binding poses and enhancing the understanding of the dynamic behavior of the protein–ligand complex. The electrostatic potential surface of the FaMAN8 protein was calculated using the Adaptive Poisson–Boltzmann Solver (APBS) [51] and visualized in Visual Molecular Dynamics (VMD) [52].

The cavity volumes and pocket areas of each protein were characterized using the CASTpFold server [53]. To capture conformational variability, seven representative frames were extracted from the first replica of the molecular dynamics trajectories for both the FaMAN8 and 6B9O structures, focusing on a single chain. These frames were individually analyzed to determine the pocket volumes and surface areas, and the results were subsequently averaged to provide a representative measurement for each protein chain. The averaged values were compiled into a table, enabling a quantitative comparison of pocket size and density across the proteins and supporting downstream interpretation of ligand accessibility and binding potential.

Molecular Mechanics Generalized Born Surface Area (MM-GBSA) calculations were performed for each protein chain across all replicates using the Prime MM-GBSA module [54] within the Schrödinger Suite. For each chain, the binding free energy between the protein and its ligand was evaluated into individual energetic contributions, including Coulombic, hydrogen bond, lipophilic, polar solvation, and van der Waals interactions. The resulting energies were compiled into a comprehensive table, providing a detailed quantitative assessment of the energetic determinants underlying protein–ligand interactions and facilitating comparative analyses across the simulations.

## 5. Conclusions

The present study offers a detailed structural and functional analysis of FaMAN8, an α-mannosidase from *F. × ananassa*, which may play a role in stress responses and plant–microbe interactions. Our findings indicate that FaMAN8 possesses a conserved domain architecture and binding pocket similar to canonical GH38 enzymes, while displaying unique expression patterns in roots subjected to combined abiotic stress and endophytic colonization. Computational modeling, molecular docking, and dynamics simulations have confirmed the stability and specificity of ligand binding, supported by favorable energetic and electrostatic interactions. The larger catalytic cavity and responsive transcriptional profile of FaMAN8 suggest a versatile enzymatic function in glycan remodeling, potentially associated with symbiotic signaling or cell wall modification under stress conditions. Collectively, these findings provide valuable insights into the molecular basis of glycosidase function in strawberry and identify FaMAN8 as a potential target for enhancing plant resilience through microbiome-assisted strategies.

## Figures and Tables

**Figure 1 ijms-26-11650-f001:**
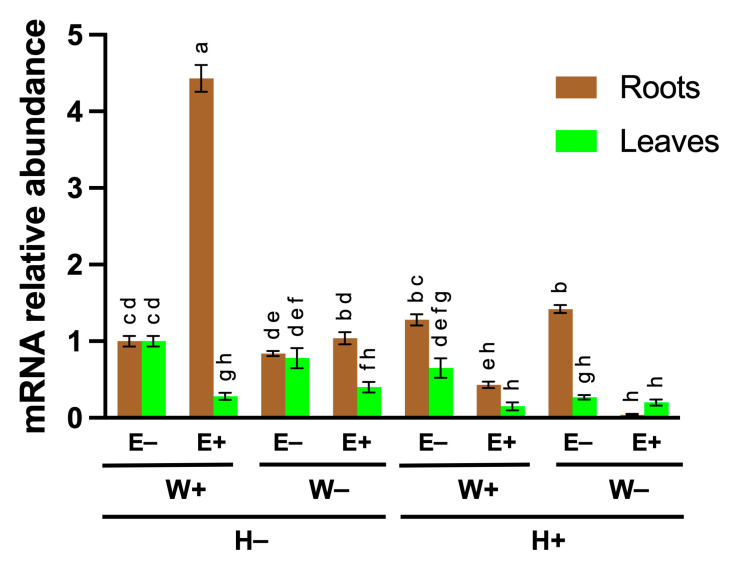
Relative expression of *FaMAN8* in roots and leaves of *Fragaria × ananassa* under different irrigation (W+: 100%; W−: 50%) and endophyte treatments (E−: without endophytes; E+: with endophytes), with or without heat stress (H+/H−). Expression was normalized against *FaGAPDH*. Bars represent means ± SD. Different letters indicate statistically significant differences (Tukey’s test, *p* < 0.05).

**Figure 2 ijms-26-11650-f002:**
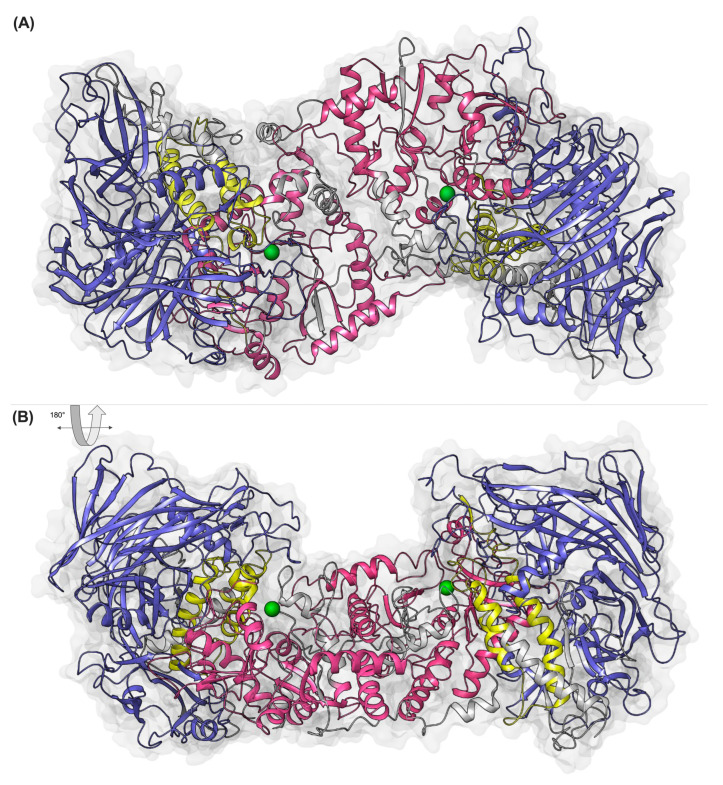
Homology model of FaMAN8 showing domain organization and structural dimerization. The N-terminal polysaccharide deacetylase-like domain is shown in pink, the central α-mannosidase domain in yellow, and the C-terminal GH38 catalytic domain in blue. Zinc ions (Zn^2+^) are depicted as green spheres. (**A**,**B**) views illustrate the overall shape and conserved fold of the dimer.

**Figure 3 ijms-26-11650-f003:**
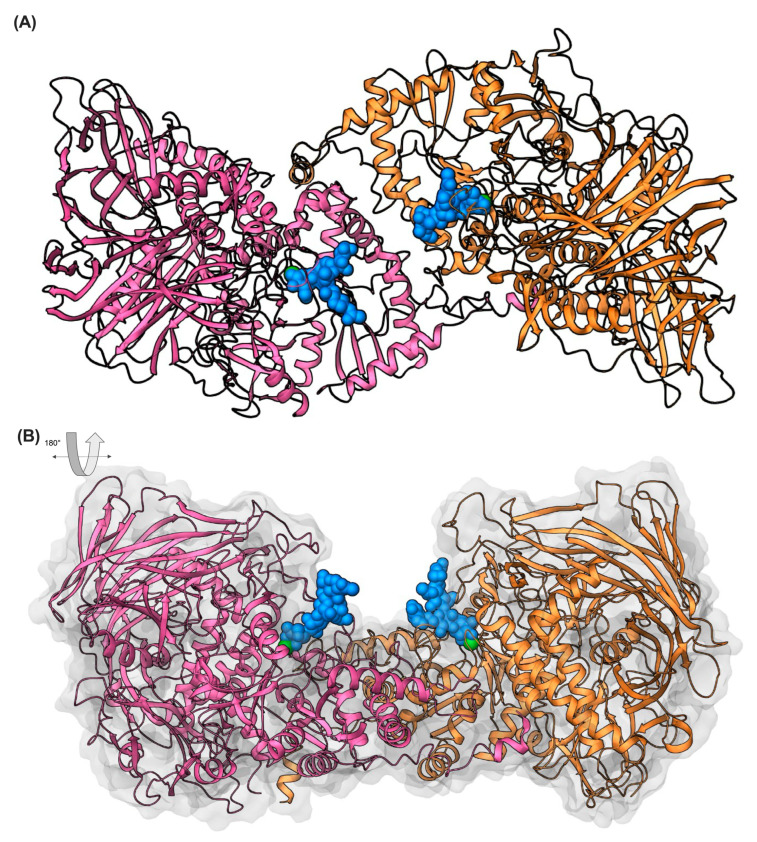
Docking simulation of FaMAN8 in complex with the glycan ligand Man_3_GlcNAc_2_ (blue spheres). The ligand fits into the conserved catalytic pocket near the Zn^2+^ ion (green spheres). Chain (**A**) and chain (**B**) of the dimer are colored pink and orange, respectively. The model suggests conserved substrate recognition features.

**Figure 4 ijms-26-11650-f004:**
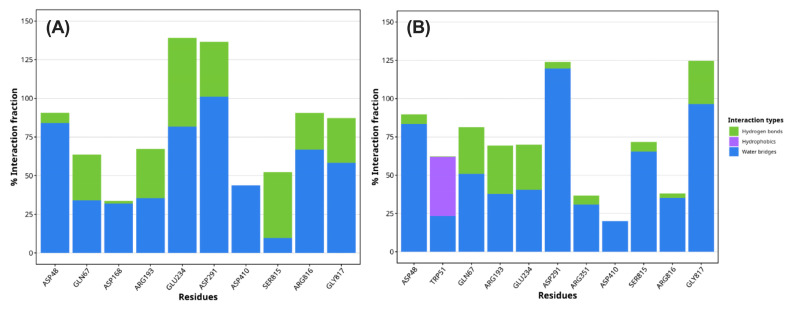
Protein–ligand interaction profiles from molecular dynamics simulations of FaMAN8. Panels show monomer (**A**) (left) and monomer (**B**) (right). Interaction types include hydrogen bonds (green), water bridges (blue), and hydrophobic contacts (purple). Residues are ranked by the percentage of simulation time spent interacting with the ligand.

**Figure 5 ijms-26-11650-f005:**
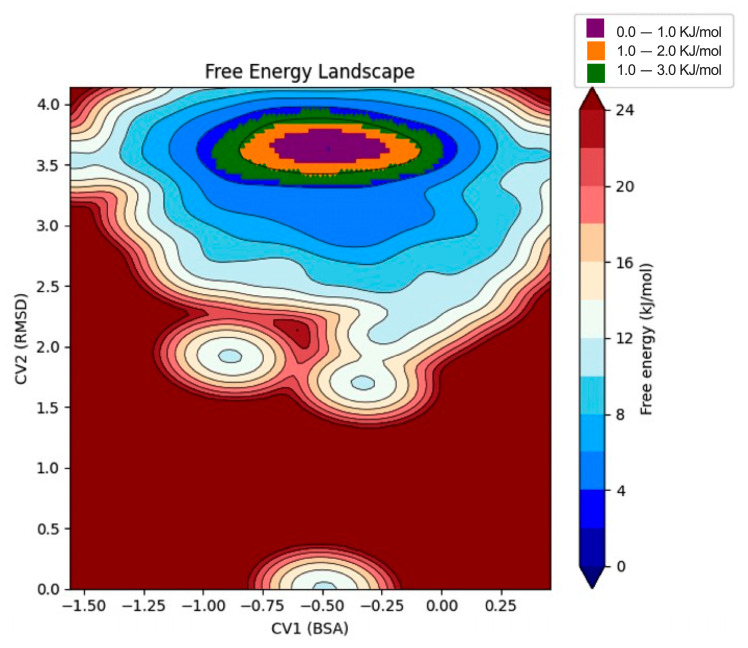
Free energy landscape (FEL) of the FaMAN8–ligand complex based on molecular dynamics simulations. RMSD and buried surface area (BSA) were used as collective variables. The main energy basin (purple to green) represents the most stable conformational state. Higher energy regions (yellow to red) indicate less favorable conformations.

**Figure 6 ijms-26-11650-f006:**
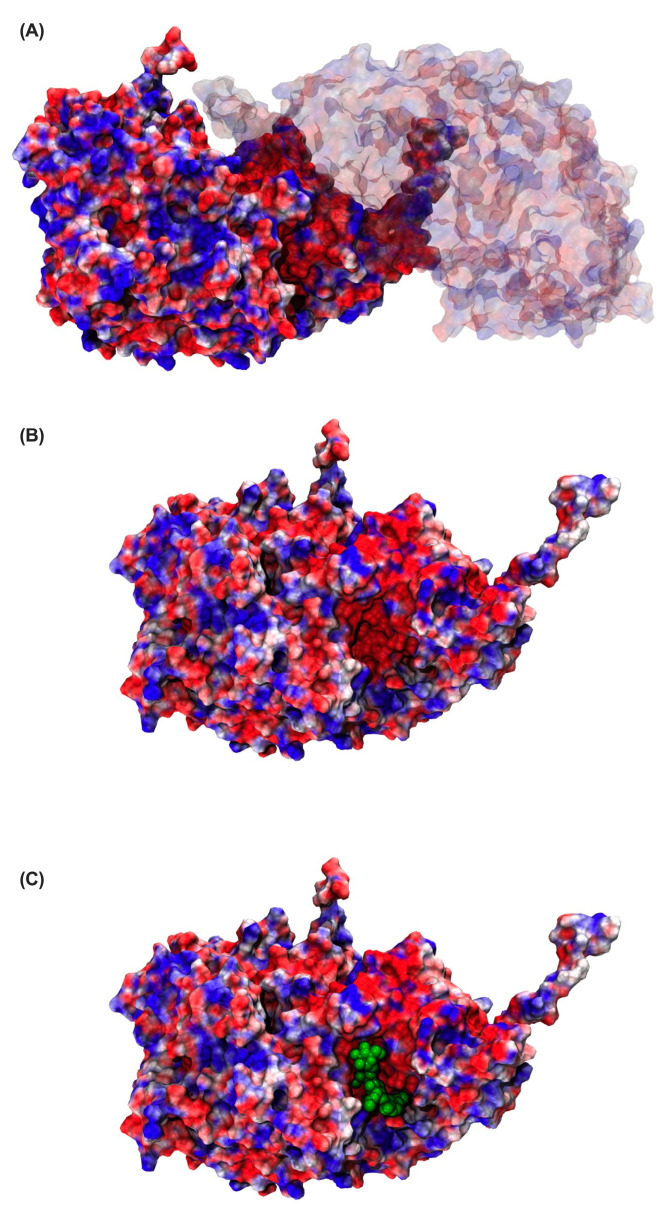
Electrostatic surface potential of FaMAN8 without (**A**,**B**) and with (**C**) the ligand. Panel (**A**) shows the full dimer, panel (**B**) focuses on one monomer with a negatively charged binding groove (red), and panel (**C**) shows the ligand (green spheres) docked into the active site. Electrostatic potentials range from negative (red) to positive (blue).

**Table 1 ijms-26-11650-t001:** Volume cavity.

Protein Cavity Volume		
Protein	Area (A^2^)	Volume (A^3^)
JbMAN	342.5 ± 192.5	211.0 ± 133.7
FaMAN8	471.4 ± 357.8	575.0 ± 791.1

**Table 2 ijms-26-11650-t002:** Molecular Mechanics/Generalized Born Surface Area (MM-GBSA) determinations.

MM-GBSA Energy Determinations	
Energy Type	FaMAN
Coulomb	−40.4 ± 17.8
Hydrogen bond	−3.9 ± 1.8
Lipophobic	−10.3 ± 4.9
Polar solvation	81.7 ± 30.9
Van del Waals	−35.8 ± 15.0

## Data Availability

The raw data supporting the conclusions of this article will be made available by the authors on request.

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
