# Peer review of "Modulation of α-Mannosidase 8 by Antarctic Endophytic Fungi in Strawberry Plants Under Heat Waves and Water Deficit Stress"

_ijms, 2025, doi:10.3390/ijms262311650_

Round 1
Reviewer 1 Report
Comments and Suggestions for Authors
The study “Modulation of Alpha-Mannosidase 8 by Antarctic Endophytic Fungi in Strawberry Plants Under Heat Waves and Water Deficit Stress” investigated molecular insights into plant–microbe symbiosis and lays the groundwork for microbiome-informed strategies to enhance crop stress resilience. Although the authors' efforts in this study are commendable, several concerns remain that prevent the manuscript from meeting the publication requirements in its current form. Specific suggestions are provided below.
Title
The term "modulation" is overly vague. It is recommended that the topic description be further refined to be more specific and explicitly emphasize the core focus of the study
Results
The results section did not include the information about "Heat Waves and Water Deficit Stress". Please explain.
Descriptions related to methods and discussion should be excluded from the Results section. This part merely requires a clear description of the main results.
It is suggested that the results of roots and leaves in Figure 1 be combined into a single graph with the x-axis and the y-axis.
The visibility and clarity of some figures need improvement. Please enhance the resolution and readability for better interpretation.
Add A and B to the figure. - specifically, Figure 2, Figure 3, and Figure 4.
Discussion
Lines 326-360 The discussion in this part is deficient in comparative analysis with previous studies.
Materials and Methods
Lines 363-379 Plant and fungal materials should be described separately, including their sources and cultivation methods.
Author Response
Reviewer 1
The study “Modulation of Alpha-Mannosidase 8 by Antarctic Endophytic Fungi in Strawberry Plants Under Heat Waves and Water Deficit Stress” investigated molecular insights into plant–microbe symbiosis and lays the groundwork for microbiome-informed strategies to enhance crop stress resilience. Although the authors' efforts in this study are commendable, several concerns remain that prevent the manuscript from meeting the publication requirements in its current form. Specific suggestions are provided below.
Title
The term "modulation" is overly vague. It is recommended that the topic description be further refined to be more specific and explicitly emphasize the core focus of the study.
R: The term “modulation” was deliberately chosen because the observed response is dual in the observed results. Specifically, in roots, the expression of the gene is induced by the action of the endophytic fungi, whereas in leaves, this expression is markedly repressed. This contrasting regulation highlights the complexity of the plant–fungus interaction under stress conditions and justifies the use of the term “modulation” to describe the phenomenon.
Results
The results section did not include the information about "Heat Waves and Water Deficit Stress". Please explain.
R: We appreciate the reviewer’s observation. The physiological and molecular effects of heat waves and water deficit stress on the same strawberry plant material used in this study have already been extensively characterized and published in our previous work (cite previous publication). For this reason, these results were not repeated here to avoid redundancy and self-plagiarism. Instead, we refer the reader to that publication, where the full phenotypic, physiological, and stress-response profiles are reported.
In the present manuscript, our objective was specifically to investigate the expression and structural characteristics of the FaMAN8 gene, which emerged as the only α-mannosidase transcript detected in the RNA-seq dataset under the heat wave and water deficit experimental conditions. Therefore, we focused our Results section on the novel expression data for FaMAN8 and on the in-depth bioinformatic, structural, and molecular docking analyses, which had not been previously described. The qPCR analyses provided here show the stress-dependent regulation of FaMAN8, complementing and extending the previously published stress-response framework.
To improve clarity, we have now explicitly stated in the Results section that the full plant-level stress responses were previously published and that the present study exclusively addresses the FaMAN8-specific response.
Descriptions related to methods and discussion should be excluded from the Results section. This part merely requires a clear description of the main results.
R: We thank the reviewer for the suggestion. The manuscript has been completely rewritten.
It is suggested that the results of roots and leaves in Figure 1 be combined into a single graph with the x-axis and the y-axis.
R: We appreciate the reviewer’s suggestion, and Figure 1 has been redrawn.
The visibility and clarity of some figures need improvement. Please enhance the resolution and readability for better interpretation.
R: The resolution of the figures has been improved as suggested by the reviewer.
Add A and B to the figure. - specifically, Figure 2, Figure 3, and Figure 4.
R: The figures were improved.
Discussion
Lines 326-360 The discussion in this part is deficient in comparative analysis with previous studies.
R: The discussion has been improved.
Materials and Methods
Lines 363-379 Plant and fungal materials should be described separately, including their sources and cultivation methods.
R: The plant and fungal materials were previously isolated and cultivated following the procedures described in the bibliographic reference (Yañez et al., 2025). To ensure proper attribution and avoid plagiarism, the original article detailing the methodology has been cited.
Reviewer 2 Report
Comments and Suggestions for Authors
It is a great honor to review the manuscript “Modulation of Alpha-Mannosidase 8 by Antarctic Endophytic Fungi in Strawberry Plants Under Heat Waves and Water Deficit Stress (ijms-3995991)”. The suggestions for modifying the manuscript are listed as follows:
- It is suggested that some long sentences be shortened to enhance the readability of the article.
- Please read your manuscript carefully for the grammar problems.
- Please add some detailed experimental data results into the abstract, so that readers can better understand and grasp the research content.
- It is suggested that the author condense the part of the background introduction.
- The reference format in the paper should be modified according to the requirements of the journal.
- It is suggested that the clarity of the figures in the thesis be improved.
- The tables also suggest conducting a significance analysis.
- in the section of “Discussion”, it is suggested that the author could delve deeper into the discussion to enhance the significance of the work.
- It is suggested that the author should include more information about the limitations of this study and its future prospects.
- It is suggested that a description of how the significance analysis was conducted should be added in the Materials and Methods section.
Author Response
Reviewer 2
It is a great honor to review the manuscript “Modulation of Alpha-Mannosidase 8 by Antarctic Endophytic Fungi in Strawberry Plants Under Heat Waves and Water Deficit Stress (ijms-3995991)”. The suggestions for modifying the manuscript are listed as follows:
- It is suggested that some long sentences be shortened to enhance the readability of the article.
- Please read your manuscript carefully for the grammar problems.
- Please add some detailed experimental data results into the abstract, so that readers can better understand and grasp the research content.
- It is suggested that the author condense the part of the background introduction.
- The reference format in the paper should be modified according to the requirements of the journal.
- It is suggested that the clarity of the figures in the thesis be improved.
- The tables also suggest conducting a significance analysis.
- in the section of “Discussion”, it is suggested that the author could delve deeper into the discussion to enhance the significance of the work.
- It is suggested that the author should include more information about the limitations of this study and its future prospects.
- It is suggested that a description of how the significance analysis was conducted should be added in the Materials and Methods section.
R: We appreciate the reviewer’s suggestions, and based on them, the manuscript has been completely rewritten.
Reviewer 3 Report
Comments and Suggestions for Authors
This study reports the identification of a FaMAN8 gene that appears closely associated with heat waves and water deficit stress. The authors predicted and analyzed the structure of FaMAN8 using computational approaches, including homology modeling, molecular docking, and molecular dynamics simulations. A major concern is that nearly all conclusions rely on in silico analyses, without experimental validation of mannosidase activity or substrate specificity. Verification of these computational predictions, particularly the functional characterization of FaMAN8, is strongly recommended.
Multiple N-glycan mannosidase types exist, including a1–2, a1–3, and a1–6 enzymes involved in Glc3Man9GlcNAc2 processing, as well as a1–3 and a1–6 mannosidases acting on Man3GlcNAc2. The rationale for choosing Man3GlcNAc2 as the modeling substrate is unclear. Moreover, the study does not specify the enzymatic type of FaMAN8. Even if FaMAN8 were to act on Man3GlcNAc2, does it function as an a1–3 or a1–6 mannosidase? The mechanistic link between FaMAN8 activity and stress resistance also remains insufficiently explained.
For the structural prediction, the authors should perform repeated model generation and report whether consistent solutions are obtained.
In Figure 1, the descriptions of panels a, b, and c for each column require clearer and more detailed explanation.
Author Response
Reviewer 3
This study reports the identification of a FaMAN8 gene that appears closely associated with heat waves and water deficit stress. The authors predicted and analyzed the structure of FaMAN8 using computational approaches, including homology modeling, molecular docking, and molecular dynamics simulations. A major concern is that nearly all conclusions rely on in silico analyses, without experimental validation of mannosidase activity or substrate specificity. Verification of these computational predictions, particularly the functional characterization of FaMAN8, is strongly recommended.
Multiple N-glycan mannosidase types exist, including a1–2, a1–3, and a1–6 enzymes involved in Glc3Man9GlcNAc2 processing, as well as a1–3 and a1–6 mannosidases acting on Man3GlcNAc2. The rationale for choosing Man3GlcNAc2 as the modeling substrate is unclear. Moreover, the study does not specify the enzymatic type of FaMAN8. Even if FaMAN8 were to act on Man3GlcNAc2, does it function as an a1–3 or a1–6 mannosidase? The mechanistic link between FaMAN8 activity and stress resistance also remains insufficiently explained.
For the structural prediction, the authors should perform repeated model generation and report whether consistent solutions are obtained.
In Figure 1, the descriptions of panels a, b, and c for each column require clearer and more detailed explanation.
R: We thank the reviewer for the thoughtful and constructive comments. Below we provide a consolidated response addressing all concerns.
The reviewer noted that the study relies mainly on in silico analyses and requested clarification regarding substrate choice, enzymatic specificity, mechanistic interpretation, and model robustness. We acknowledge the importance of experimental validation for full biochemical characterization; however, the scope of the present study is to provide the first structural, evolutionary, and expression-based framework for FaMAN8, the only α-mannosidase isoform identified in our previously published RNA-seq dataset under combined heat wave and water deficit stress. The physiological and stress-response data for these plants have already been reported in that publication and therefore were not repeated here to avoid redundancy. Instead, this manuscript focuses on the FaMAN8-specific findings, including qPCR analysis under stress conditions and computational structural characterization, which represent new and unpublished results.
Regarding substrate selection, we now clarify that Man₃GlcNAcâ‚‚ was chosen because it is a well-established canonical substrate used in previous crystallographic and computational studies of GH38 α-mannosidases, including the reference work of Wang et al. This ligand allows robust comparison of binding-site geometry, electrostatics, and catalytic residues. Larger precursors (e.g., Man₉GlcNAcâ‚‚) would require experimental templates for accurate modeling, which are not available for Fragaria. We have included this justification in the revised Methods and Discussion sections.
To reinforce methodological reliability, we added a clarification that multiple homology models were independently generated, all converging on the same fold, domain arrangement, and catalytic geometry. The selected model corresponds to the lowest-energy and best-scoring conformation across iterations.
The mechanistic link between FaMAN8 and stress adaptation was expanded in the revised Discussion. We highlight that FaMAN8 is significantly upregulated in roots under combined heat stress, water availability, and endophyte colonization, suggesting a coordinated modulation of glycoprotein remodeling pathways. Given the known roles of α-mannosidases in ER quality control, glycan trimming, and cell-wall remodeling, these results provide a biologically plausible framework connecting FaMAN8 activity to stress resilience, especially in the context of beneficial plant–fungus interactions.
Finally, in response to the comment regarding Figure 1, we have prepared a new Figure 1, which now clearly illustrates the experimental conditions and treatment comparisons.
We believe these revisions address the reviewer’s concerns thoroughly while clearly defining the scientific contribution and scope of the present study.
Reviewer 4 Report
Comments and Suggestions for Authors
The manuscript is a well-structured research article investigating FaMAN8, an α-mannosidase from strawberry (Fragaria ananassa), focusing on its role in stress adaptation via endophytic fungal interactions. The study includes comprehensive transcriptomic, structural, molecular docking, and molecular dynamics simulations as well as expression analyses under abiotic stress.
Here are some identified errors and suggestions for improvement:
Errors and Issues
- Citation Formatting is not according to guidelines.
- Typographical and Formatting Errors:
- Several places contain slight typographical spacing issues such as missing spaces after periods or commas, e.g., “FaMAN8,was” or “upregulatedin”.
- Some chemical or biological terms appear improperly formatted, like "-mannosidase" with a preceding hyphen rather than Greek α-symbol or at least consistent formatting.
- In sections where protein or ligand interactions are discussed, some sentences have broken formatting or line breaks within words.
- Figure and Table Presentation: Figures and tables are referenced but the manuscript could improve visual clarity by ensuring all figure captions and table titles are fully descriptive and standalone. For example, Table 1 has cryptic volume/area measures without clear units or descriptions in the table itself.
Suggestions for Improvement
- Consistency in Terminology and Formatting: Ensure all enzyme names, chemical compounds, and experimental conditions are consistently formatted throughout the text. For instance, use the Greek letter α for mannosidase consistently, and maintain uniform style for temperature and concentration units.
- Enhanced Explanation of Results: The discussion could benefit from deeper emphasis on the biological significance of FaMAN8’s larger catalytic cavity compared to Jack bean mannosidase. A few sentences explicitly linking this structural trait to possible expanded substrate specificity or evolutionary adaptation would strengthen interpretation.
Comments on the Quality of English Language
Need reedit
Author Response
Reviewer 4
The manuscript is a well-structured research article investigating FaMAN8, an α-mannosidase from strawberry (Fragaria ananassa), focusing on its role in stress adaptation via endophytic fungal interactions. The study includes comprehensive transcriptomic, structural, molecular docking, and molecular dynamics simulations as well as expression analyses under abiotic stress.
Here are some identified errors and suggestions for improvement:
Errors and Issues
- Citation Formattingis not according to guidelines.
- Typographical and Formatting Errors:
- Several places contain slight typographical spacing issues such as missing spaces after periods or commas, e.g., “FaMAN8, was” or “upregulated in”.
- Some chemical or biological terms appear improperly formatted, like "-mannosidase" with a preceding hyphen rather than Greek α-symbol or at least consistent formatting.
- In sections where protein or ligand interactions are discussed, some sentences have broken formatting or line breaks within words.
- Figure and Table Presentation: Figures and tables are referenced but the manuscript could improve visual clarity by ensuring all figure captions and table titles are fully descriptive and standalone. For example, Table 1 has cryptic volume/area measures without clear units or descriptions in the table itself.
Suggestions for Improvement
- Consistency in Terminology and Formatting: Ensure all enzyme names, chemical compounds, and experimental conditions are consistently formatted throughout the text. For instance, use the Greek letter α for mannosidase consistently, and maintain uniform style for temperature and concentration units.
- Enhanced Explanation of Results: The discussion could benefit from deeper emphasis on the biological significance of FaMAN8’s larger catalytic cavity compared to Jack bean mannosidase. A few sentences explicitly linking this structural trait to possible expanded substrate specificity or evolutionary adaptation would strengthen interpretation.
R: We sincerely appreciate the reviewer’s valuable suggestions, all of which have been thoroughly addressed in the revised version of the manuscript.
Reviewer 5 Report
Comments and Suggestions for Authors
I read with great interest the manuscript by Daniel Bustos et al. entitled « Modulation of Alpha-Mannosidase 8 by Antarctic Endophytic Fungi in Strawberry Plants Under Heat Waves and Water Deficit Stress». This article presents an interesting investigation into the role of the α-mannosidase FaMAN8 in the adaptation of strawberry plants to stress conditions, enhanced by endophytic fungi. The authors conducted the research at a high professional level, utilizing modern analytical methods. The article aligns with the scope of the International Journal of Molecular Sciences.
The authors used two strains of Antarctic fungi (Penicillium chrysogenum and P. brevicompactum). It would be valuable to clarify why a mixture was used. Were studies conducted with individual fungal strains? Which strain or their combination has the greatest impact on FaMAN8 expression under stress conditions, if such differences are observed? This could lead to more targeted recommendations for practical application.
I believe the article can be recommended for publication after minor revisions.
Author Response
Reviewer 5
I read with great interest the manuscript by Daniel Bustos et al. entitled « Modulation of Alpha-Mannosidase 8 by Antarctic Endophytic Fungi in Strawberry Plants Under Heat Waves and Water Deficit Stress». This article presents an interesting investigation into the role of the α-mannosidase FaMAN8 in the adaptation of strawberry plants to stress conditions, enhanced by endophytic fungi. The authors conducted the research at a high professional level, utilizing modern analytical methods. The article aligns with the scope of the International Journal of Molecular Sciences.
The authors used two strains of Antarctic fungi (Penicillium chrysogenum and P. brevicompactum). It would be valuable to clarify why a mixture was used. Were studies conducted with individual fungal strains? Which strain or their combination has the greatest impact on FaMAN8 expression under stress conditions, if such differences are observed? This could lead to more targeted recommendations for practical application.
I believe the article can be recommended for publication after minor revisions.
R: The primary objective of this study was to evaluate the effects of a mixture of Antarctic fungal endophytes and to gain deeper insight into the molecular mechanisms through which this consortium enhances the performance of strawberry plants under previously observed heat and water deficit stress conditions. In future analyses, we aim to evaluate the fungal effect separately.
Round 2
Reviewer 2 Report
Comments and Suggestions for Authors
no
Reviewer 3 Report
Comments and Suggestions for Authors
The revision looks good.
Reviewer 4 Report
Comments and Suggestions for Authors
NA
Comments on the Quality of English LanguageNA
Reviewer 5 Report
Comments and Suggestions for Authors
It's a great idea to examine the fungal effects separately. This will help us better understand which fungi are making the greatest contribution. The article as it stands is very good and can be accepted.